# High-quality mouse reference genomes reveal the structural complexity of the murine protein-coding landscape

## Graphical abstract

## Authors

Mohab Helmy, Jin U. Li, Xinyu F. Yan, ..., Clare M. Smith, Jingtao Lilue, Thomas M. Keane

## Correspondence

tk2@ebi.ac.uk

## In brief

Helmy et al. provide a collection of high-quality mouse reference genomes. They were able to resolve some of the most complex regions among mouse genomes that are involved in immune defense. These findings have an impact on a variety of mouse genetics experiments as well as the usage of mice as an animal model for medical research.

## Highlights

- Collection of high-quality mouse reference genomes

- Insights into the structural complexity of key regions of the mouse genome

- Impact of using a strain-specific genome for RNA-seq analysis

Helmy et al., 2026, Cell Genomics 6, 101074
February 11, 2026 © 2025 The Author(s). Published by Elsevier Inc.

CellPress

## Article

# High-quality mouse reference genomes reveal the structural complexity of the murine protein-coding landscape

Mohab Helmy,[1,2,16] Jin U. Li,[3,16] Xinyu F. Yan,[3,16] Rachel K. Meade,[8,14,16] Elizabeth Anderson,[4] Patrick B. Chen,[5] Anne M. Czechanski,[9] Tomás Di Domenico,[2,12] Jonathan Flint,[5] Erik Garrison,[6] Marco T.P. Gontijo,[8] Andrea Guarracino,[6] Leanne Haggerty,[1] Edith Heard,[7] Kerstin Howe,[4] Narendra Meena,[2] Fergal J. Martin,[1] Eric A. Miska,[2,13] Isabell Rall,[7] Navin B. Ramakrishna,[2] Alexandra Sapetschnig,[2] Swati Sinha,[1] Diandian Sun,[3,11] Francesca F. Tricomi,[1] Runjia Qu,[3] Jonathan M.D. Wood,[4] Tianzhen Wu,[3] Dian J. Zhou,[3] Laura Reinholdt,[9] David J. Adams,[4] Clare M. Smith,[8,14,15] Jingtao Lilue,[3,15] and Thomas M. Keane[1,10,15,17,*]

[1]European Molecular Biology Laboratory, European Bioinformatics Institute, Wellcome Genome Campus, Hinxton, Cambridge CB10 1SD, UK
[2]Department of Biochemistry, Department of Genetics and Gurdon Institute, University of Cambridge, Cambridge, UK
[3]Oujiang Laboratory, Wenzhou Medical University, Wenzhou, China
[4]Wellcome Sanger Institute, Wellcome Genome Campus, Hinxton CB10 1SA, UK
[5]Brain Research Institute, University of California, Los Angeles, Los Angeles, CA, USA
[6]University of Tennessee Health Science Center, Memphis, TN, USA
[7]European Molecular Biology Laboratory (EMBL), Heidelberg, Germany
[8]Department of Molecular Genetics and Microbiology, Duke University, Durham, NC, USA
[9]The Jackson Laboratory, Bar Harbor, ME, USA
[10]University of Nottingham, Nottingham, UK
[11]Central South University, Changsha, Hunan Province, China
[12]Bioinformatics Unit, Spanish National Cancer Research Centre (CNIO), Madrid, Spain
[13]Wellcome Sanger Institute, Wellcome Genome Campus, Hinxton CB10 1SA, UK
[14]University Program in Genetics and Genomics, Duke University, Durham, NC, USA
[15]Senior author
[16]These authors contributed equally
[17]Lead contact
*Correspondence: tk2@ebi.ac.uk

## SUMMARY

We present a collection of 17 high-quality long-read inbred mouse strain genomes with complete annotation (contig N50s of 0.8–33.9 Mbp). This collection includes 12 widely used classical laboratory strains and 5 wild-derived strains. We have resolved previously incomplete genomic regions, including the major histocompatibility complex (MHC), defensin cluster, T cell receptor, and Ly49 complexes. Hundreds of non-reference genes from previous publications not found in GRCm39, such as *Defa1*, *Raet1a*, and *Klra20* (*Ly49T*), were localized in the new reference genomes. We conducted a genome-wide scan of variable number tandem repeats (VNTRs) within the coding regions, identifying over 400 genes with VNTR polymorphisms with up to 600 repeat copies and repeat units reaching 990 nucleotides. Our strain-specific annotations enhance RNA sequencing (RNA-seq) analyses, as demonstrated in PWK/PhJ, where we observed a 5.1% improvement in read mapping and expression-level differences in 2.1% of coding genes compared to using GRCm39.

## INTRODUCTION

The mouse has long been the foremost mammalian model for studying human disease and human health. Since the first inbred mouse strain, DBA, was established in 1909 by Dr. C.C. Little,[1] more than 450 inbred strains have been developed.[2] In 2002, the mouse reference genome was based on a single inbred mouse strain, namely, C57BL/6J (BL6).[3] The completion of the mouse reference genome has facilitated the manipulation of mouse genes, as well as the development of other molecular

tools for biomedical research. However, the availability of just one high-quality reference genome for only the BL6 strain significantly biased the usage of mouse strains in laboratories: over time, research shifted toward predominant use of the BL6 strain. According to PubMed, the application of BL6-related strains has increased from ∼25% in the 1980s to more than 70% in 2024 (Figure S1). However, non-BL6 mouse strains have provided valuable resources and insights for biomedical research, which a complete sequence will augment. For example, embryonic stem (ES) cell lines from the 129 mouse strain proved most

suitable for culture manipulation and repopulation of the germline,[4] resulting in many knockout mouse strains on a 129 background; DBA/2J has broad developmental, neurobiological, and immunological differences from BL6, including resistance to SARS-CoV-2[5]; BALB/c exhibit greater resistance to roundworms than BL6[6]; and a wide range of wild-derived mouse strains are resistant to virulent *T. gondii*,[7,8] lipopolysaccharide (LPS)-induced lethality,[9] and cerebral malaria infection.[10]

Since the early 2010s, several hybrid mouse panels have been developed to leverage genetic and phenotypic diversity in mouse subspecies, namely the collaborative cross (CC),[11] the diversity outbred (DO) population,[12] and the C57BL/6J x DBA/2J (BXD) recombinant inbred panel.[13] In parallel, comprehensive genomic variation catalogs (SNPs, insertions or deletions [indels], tandem repeats, and structural variants) from dozens of strains[14–17] and *de novo* assemblies of 16 key mouse strains have highlighted the extent of "high-diversity" loci in the mouse genome,[18] revealing novel genes absent in the BL6 reference genome and mitochondrial genome variation.[18] These genes are confirmed to play key roles in immunity, behavior, sensory functions, and reproductive phenotypes. Of note, these initial assemblies, produced using short-read sequencing, are incomplete in many complex and repeat-rich regions.

In this paper, we present a collection of high-quality long-read assemblies of 17 widely used laboratory mouse strains and wild-derived inbred strains (Figure 1). These genome assemblies achieve comparable sequence and gene annotation quality to the GRm39 reference genome. We highlight how these genomes will enhance our understanding of mouse genome architecture and complexity, improve genomic analysis applications, and accelerate genetic association studies toward causal variant identification. These resources will address the historical strain bias in mouse studies and facilitate future biomedical research with non-BL6 mice.

## RESULTS

### Reference-quality chromosome-scale mouse genomes

Chromosome-scale *de novo* assemblies of 17 mouse strains were generated using a combination of PacBio continuous long reads (CLRs) and HiFi sequencing, and Hi-C for chromosome scaffolding (Tables 1 and S1). The genomes underwent several rounds of manual curation, polishing, and base error correction, yielding assemblies with a total length for chromosomes 1–19 and X (excluding unknown bases, e.g., Ns) between 2.46 and 2.68 Gbp, compared to 2.56 Gbp in GRCm39. The contig N50s are lower than those of GRCm39 (0.82–33.9 Mbp vs. 57.4 Mbp for GRCm39); however, the scaffold N50s are comparable to those of GRCm39 for several of the genomes (11–116 Mbp vs. 100.9 Mbp for GRCm39). Whole-genome sequencing (WGS) reads from each strain were used to calculate the consensus base quality (QV), with assembly QV scores ranging between Q40 and Q52.18 (99.99%–99.9993% accuracy). The assembly base error rate, estimated by aligning short reads from each strain to its respective assembly and calling SNPs, ranged from 0.38 to 2.45 SNPs per 1 Mbp, compared to 2.07 for GRCm39. Pan-metazoan BUSCO gene content completeness ranged from 99.10% to 99.60% across assemblies, comparable to GRCm39's 99.60%.

Approximately 3.55–130.77 Mbp of sequence is unplaced per strain, and gaps consist of 0.05–41.61 Mbp of unknown bases. The genomes produced by HiFi+Hi-C (DBA/2J, LP/J, NZO/HILtJ, and JF1/MsJ) have the lowest fraction of unknown bases and the longest contigs (Table S1). The total repeat content of the genomes is comparable to GRCm39 for short interspersed nuclear elements (SINEs) (96.22%–101.33%) and slightly reduced for long interspersed nuclear elements (LINEs) (89.02%–104.08%) and endogenous retroviral elements (ERVs) (86.96%–103.79%).

In 2019, we determined the most genetically diverse regions of the mouse genome, where genetic variation can exceed interspecies variation.[18,19] These regions are enriched for infection immunity and sensory gene families and transposons and thus are challenging for genome assembly. In our current assemblies, these regions are highly contiguous (96%–100% are spanned by less than three contigs; Table S2).

To assess the impact of using these long-read assemblies as reference genomes, we aligned WGS short reads from the 17 strains to three references: GRCm39, short-read assemblies,[18] and our long-read assemblies. We found improvement across several alignment metrics (Figure 1B; Table S3) when using long-read assemblies compared to GRCm39 or the short-read sequencing assemblies.[18] For instance, among the wild-derived strains, the fraction of reads where pairs align to different chromosomes varies between 0.58% and 3.00% using the long-read assemblies compared to 1.24%–4.67% and 7.93%–9.88% while using GRCm39 or the short-read assemblies, respectively.

Genome annotation was carried out using a combination of lifting over GRCm39 genes and strain-specific RNA sequencing (RNA-seq) data (see STAR Methods and Table S4). The total number of protein-coding genes ranges from 21,592 to 23,187 across the strains.

### Sequence diversity in the mouse pangenome

We used minigraph to build a mouse pangenome of 18 inbred mouse strains, including BL6. In this graph, GRCm39 serves as the backbone, with non-reference sequences represented as branches in the graph (Figure 1A). Since the pangenome graph fully represents the genomes of the strains, we can use it to interrogate the complete set of non-reference sequences and haplotypes. By traversing the pangenome graph, we identified all non-BL6 paths and merged these into the most divergent non-reference loci (see STAR Methods). Figure 1A shows the amount of non-reference sequence across each chromosome for the wild-derived strains. Non-reference loci that are shared between wild-derived strains are visible, as well as loci that are unique to one or more strains.

We used minigraph's pangenome graph to quantify the non-reference divergence of each strain compared to GRCm39 (see STAR Methods). Figure 1C shows the cumulative size of these non-reference regions per chromosome, which varies between 1.26 Kbp per 1 Mbp in chromosome 16 of C57BL/6NJ and 175.82 Kbp per 1 Mbp of chromosome X in SPRET/EiJ.[20] Among classical mouse strains, the non-reference sequence per chromosome spans from 12.8 to 58.57 Kbp per Mbp, with a median of 30.3 Kbp per Mbp.

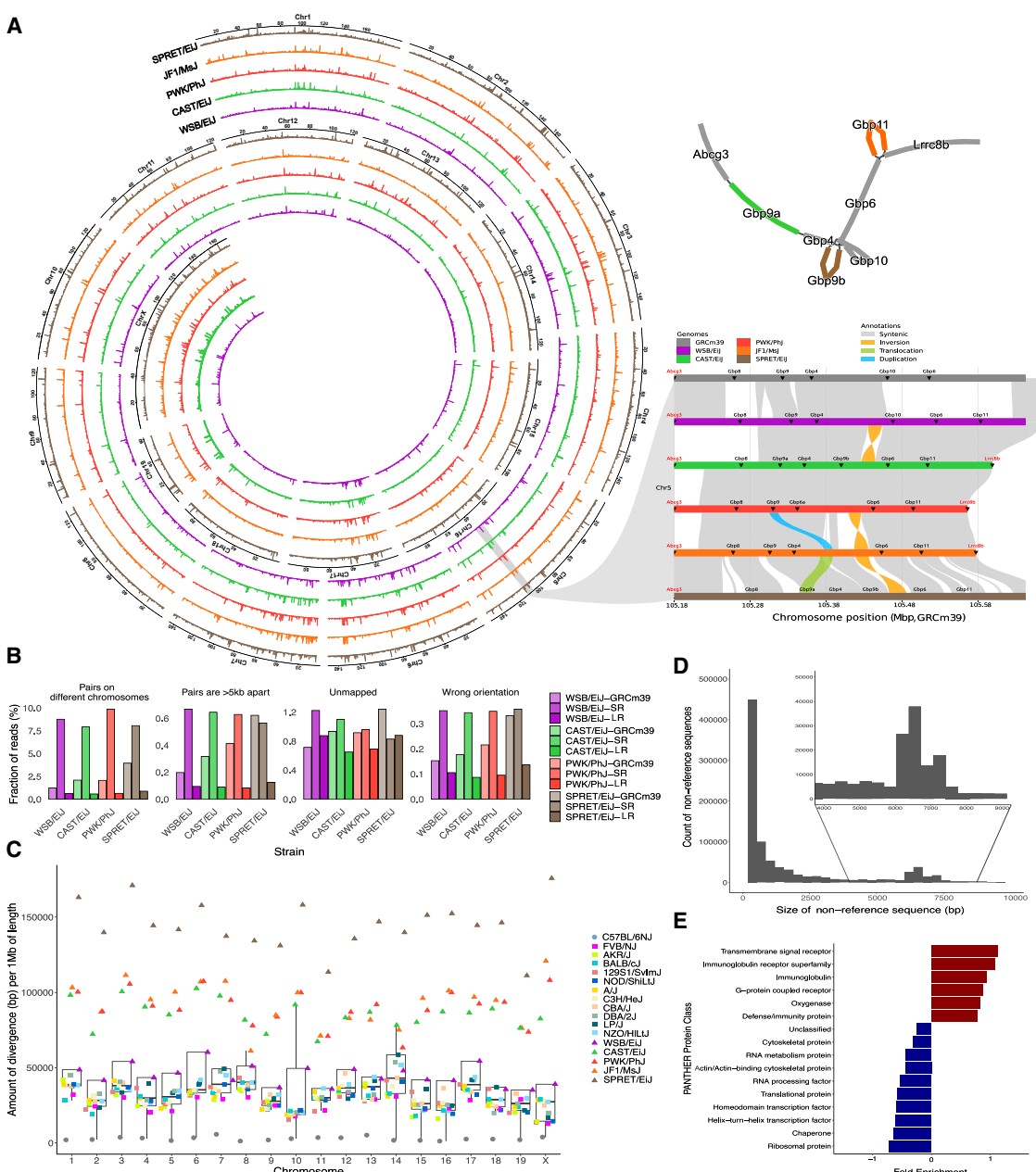

**Figure 1. The mouse pangenome produced from 17 high-quality long-read reference genomes**

(A) Pangenome sequence diversity of the wild-derived strains SPRET/EiJ, JF1/MsJ, PWK/PhJ, CAST/EiJ, and WSB/EiJ relative to the GRCm39 reference genome across autosomes and chromosome X (y axis represents the amount of non-reference sequence) (left). The pangenome graph (top) and linear synteny representation (bottom) of the GBP locus on chromosome 5 show the structural diversity (insertions, duplications, and translocations) across the locus among different strains (right).

(B) Comparison of WGS short-read mapping rates for the wild-derived strains when using GRCm39, draft short-read assemblies,[18] and these high-quality long-read assemblies.

(C) For each chromosome, the non-reference sequence from the minigraph pangenome per 1 Mbp of chromosome length across all strains.

(D) Sizes of the non-reference regions from the pangenome graph across all strains, highlighting the enrichment of full-length LINEs (~6–7 Kbp).

(E) PANTHER protein class enrichment of genes within the diverse regions of the pangenome.

The non-reference regions in the pangenome vary in size between 50 bp (default minimum threshold of minigraph) and 207.2 Kbp. Notably, the size distribution shows an enrichment at 6–7 Kbp, reflecting the size of full-length LINEs (Figure 1D). The loci of highest divergence (top 5%) contained between 101 and 3,288 protein-coding genes in C57BL/6NJ and

**Table 1. Summary statistics of the 17 mouse strain long-read assemblies for the autosomes (chromosomes 1–19) and chromosome X**

| Assembly | | | | Annotation | Repeat annotation (as % of GRCm39) | | | Gene content |
|---|---|---|---|---|---|---|---|---|
| Strain | Length (Gbp) | Contig N50 (Mbp) | Scaffold N50 (Mbp) | Protein-coding genes | SINE | LINE | ERV | BUSCO |
| C57BL/6NJ | 2.5 | 5.31 | 57.35 | 22,028 | 97.60 | 92.15 | 89.09 | 99.50% |
| FVB/NJ | 2.46 | 0.82 | 11.17 | 21,832 | 97.08 | 89.10 | 87.79 | 99.50% |
| AKR/J | 2.49 | 2.08 | 61.79 | 21,908 | 97.42 | 91.53 | 88.65 | 99.50% |
| BALB/cJ | 2.51 | 23.81 | 100.97 | 22,076 | 97.81 | 92.19 | 89.57 | 99.60% |
| 129S1/SvlmJ | 2.5 | 3.96 | 61.79 | 21,900 | 97.53 | 92.10 | 88.99 | 99.50% |
| NOD/ShiLtJ | 2.5 | 4.33 | 87.25 | 21,978 | 97.51 | 92.09 | 88.86 | 99.50% |
| A/J | 2.51 | 9.24 | 118.92 | 22,058 | 97.74 | 92.25 | 89.57 | 99.60% |
| C3H/HeJ | 2.5 | 7.87 | 74.56 | 21,980 | 97.55 | 92.18 | 89.20 | 99.50% |
| CBA/J | 2.5 | 8.74 | 114.12 | 21,968 | 97.65 | 92.12 | 89.10 | 99.60% |
| DBA/2J | 2.57 | 30.78 | 118.6 | 22,753 | 99.63 | 95.69 | 93.63 | 99.50% |
| LP/J | 2.64 | 18.07 | 129.2 | 22,642 | 100.36 | 97.69 | 96.34 | 99.50% |
| NZO/HILtJ | 2.68 | 33.89 | 93.7 | 22,652 | 100.10 | 97.28 | 95.99 | 99.60% |
| WSB/EiJ | 2.48 | 1.32 | 85.73 | 21,814 | 97.27 | 90.73 | 88.37 | 99.50% |
| CAST/EiJ | 2.49 | 22.55 | 107.57 | 21,904 | 97.44 | 89.02 | 88.59 | 99.20% |
| PWK/PhJ | 2.5 | 9.58 | 91.55 | 21,911 | 97.68 | 90.44 | 88.30 | 99.20% |
| JF1/MsJ | 2.62 | 14.97 | 55.6 | 23,187 | 101.33 | 104.08 | 103.79 | 99.10% |
| SPRET/EiJ | 2.53 | 3.5 | 116.37 | 21,592 | 96.22 | 98.29 | 86.96 | 99.40% |
| GRCm39 | 2.56 | 57.4 | 100.9 | 21,885 | – | – | – | 99.60% |

SPRET/EiJ, respectively, and 419–614 protein-coding genes among classical strains (Table S5). We examined the protein classes of these genes using PANTHER, and the results show an enrichment for defense and immunity protein classes, highlighting the biomedical importance of such loci (Figure 1E).

Among the loci of highest divergence in the pangenome (Figure 1A) is a region on chromosome 5 (105.08–105.64 Mbp in GRCm39) that encodes guanylate-binding proteins (GBPs), a group of large GTPases that play essential roles in immune responses, cell signaling, and host defense against pathogens. These proteins are crucial for interferon-induced responses, helping to combat intracellular bacterial, viral, and parasitic infections.[21] Evolutionary gain and loss of Gbp family members have shaped the diverse immune responses in different mouse strains.[22,23] Wild-derived strains show extensive diversity, with locus size varying greatly (0.39–0.48 Mbp) due to structural variants (including inversions, translocations, and duplications). In addition, different combinations of Gbp alleles are present among the strains, with some strains missing alleles that are present in BL6 and others harboring novel alleles that were annotated using *de novo* RNA-seq data in the wild-derived strains.

### Mouse MHC pangenome

The major histocompatibility complex (MHC), a core component of the vertebrate immune system, has challenged immunologists, geneticists, and evolutionary biologists for over half a century.[24] The MHC encodes genes critical for self-/non-self-recognition, restricted antigen presentation, and other important roles in immunity,[25] autoimmunity,[26] or even sexual mate selection.[27] The mouse MHC *H2* locus on chr17 is one of the most polymorphic regions in the mouse genome. For easier recognition, these polymorphic alleles are denominated with single letters, e.g., *b* for BL6 and *k* for AKR.[28] In 2024, the first mouse telomere-to-telomere (T2T) assemblies contained the complete H2 locus for BL6 and CAST/EiJ,[29,30] but most other important H2 haplotypes remain incomplete.

Our 17 genome assemblies have resolved *H2* haplotypes *a*, *bc*, *d*, *k*, *q*, *g7*, and *z* in the classical laboratory mouse strains, as well as the H2(*b*) in C57BL/6NJ. Figure 2A shows the fine structure and sequence haplotype from gene *Tapbp* to *Trim26* for all 12 laboratory mouse strains. Gaps in *H2-Q* and *H2-T* loci of the GRCm39 reference genome have been filled. Comparing the completed H2 loci between mouse strains reveals different levels of polymorphism. For H2-K and classical class II MHC molecules, gene order and structure are highly conserved. However, the high sequence diversity within protein-coding sequence (CDS) and a high dN/dS value indicate directional selection (positive selection), as previously reported.[31,32] For other class Ia members, however, the co-linearity between mouse genomes is significantly disrupted by gene recombinations, with presence/absence polymorphisms and gene copy-number variations being the primary diversity (Figure 2A, bottom).

To confirm the accuracy of our assembly, we validated it with plasmid and fosmid results.[33–38] Our *de novo* assembly agrees with the previously published genome structure of all *H2* loci (Figures 2A, S2, and S3). As a result, we labeled the *H2-D/L/Q* genes in haplotypes *k*, *q* (partial), and *d* according to previous publications. In haplotypes *q* and *z*, we identified four novel *H2-D* homologs (D5, D6, D7, and D8) and three *H2-Q* (Q16, Q17, and Q18). These genes are named according to their location on the genome and phylogenetic tree (Figure 2B). From our phylogenetic analysis, there is no clear separation between

CellPress

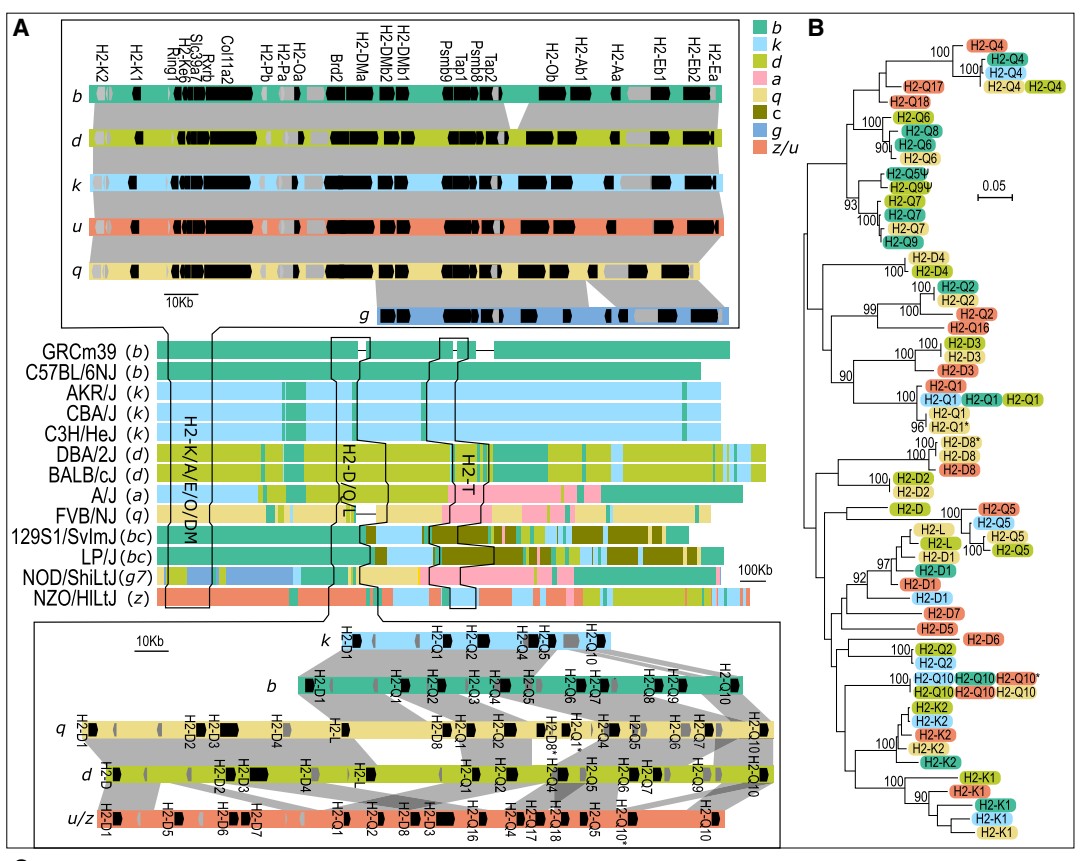

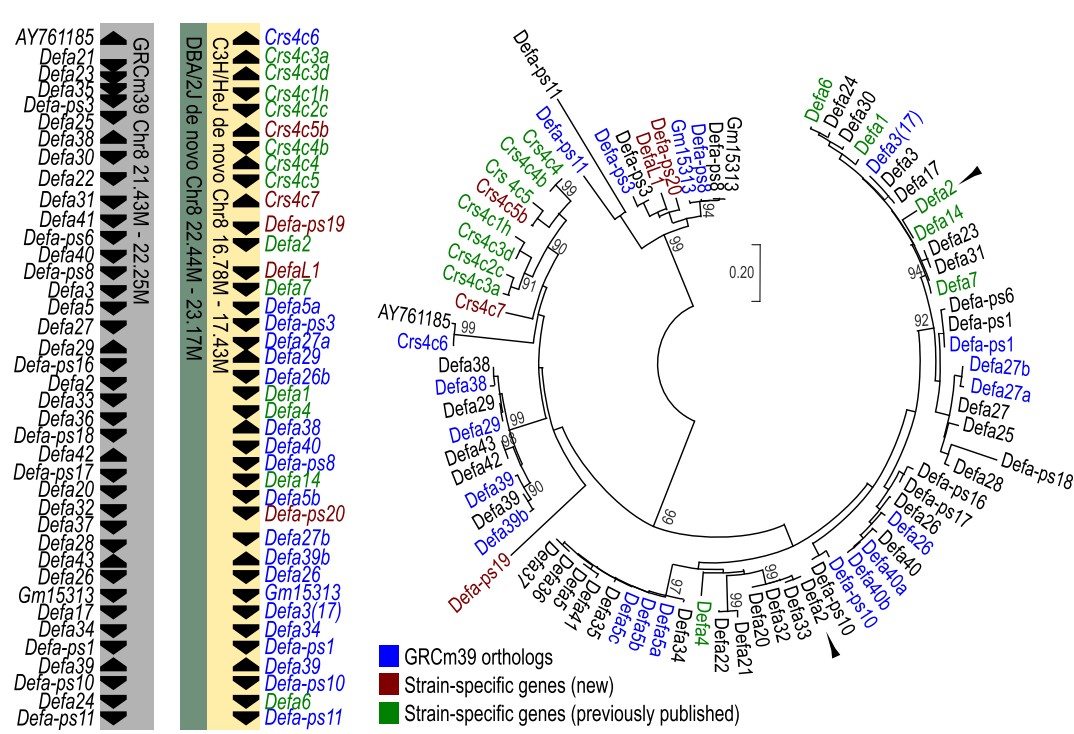

H2-D, H2-L, and H2-Q members, indicating a complex history of recombination in this locus. There was a gap in the *H2-D/Q* locus of haplotype *q* assembly, which was subsequently curated with assembly scaffolds and raw reads (Data S1). A similar polymorphism can also be found in the *H2-T* locus (Figure S3).

We performed the same analysis on five wild-derived strains (WSB/EiJ, PWK/PhJ, JF1/MsJ, CAST/EiJ, and SPRET/EiJ), identifying five additional haplotypes that are distinct from any known laboratory mouse *H2* haplotype (see STAR Methods). Notably, none of these wild-derived strains share the same *H2* haplotypes with laboratory strains or with each other (Figure S4).

### Rediscovery of non-reference genes

Prior to the mouse reference genome, BL6 was not the most popular mouse strain for biomedical research (Figure S1). Historically, most research was performed using BALB, 129, C3H, DBA, CBA, and FVB lines. Consequently, genes cloned or sequenced from non-BL6 mice are often absent from the reference genome. We downloaded the Entrez gene markers from the Mouse Genome Informatics (MGI) database (https://www.informatics.jax.org/downloads/reports/MGI_EntrezGene.rpt) and identified 278 gene markers absent from GRCm39 ("non-withdrawn" and without "genome coordinate"). We extracted sequences for 196 from public databases or publications and aligned them to both GRCm39 and our *de novo* genome assemblies (Table S6). The average sequence similarity was only 95.49% to GRCm39 but reached 99.41% to the appropriate strain genome assembly (Figure S5). Within these 196 genes, 124 genes perfectly match our assemblies, while only 12 genes are not found in the strain reference genomes, likely due to mutations during cloning, sequencing errors, or differences between sub-strains of mice (e.g., *Defa8*; see below). α-Defensin (cryptdin) members serve as an example (Figure 2C). Paneth cell α-defensins shape intestinal microbiota composition, which influences multiple biomedical processes of the host, including behavior,[39] autoimmune disease,[40] and cancer.[41] The GRCm39 reference genome lacks *Defa1*, *Defa4*, and *Defa6-16*, which were previously cloned from mouse strains C3H, 129, or DBA[42–44] and have key roles in immunity.[45–47] There are homologs of many published cryptdin-related sequence (CRS) peptides[42] that are absent in the GRCm39 reference genome. We have analyzed the α-defensin locus within the *de novo* assemblies of DBA/2J and C3H/HeJ (Figure 2C), which share the same haplotype of α-defensin. In the GRCm39 reference, 39 *Defa* members (including 9 pseudogenes) are encoded contiguously in a ~800 Kbp locus at chr8:21.4–22.2 Mbp on GRCm39. The DBA/2J genome contains the same number of *Defa* coding units. However, only 19 genes have direct orthologs in the reference (shown in blue). We identified 13 genes previously sequenced and published that are absent in GRCm39

(shown in green), including *Defa1*, *Defa4*, *Defa6*, *Defa7*, and *Defa14*. Notably, *Defa2* in the reference genome is not the ortholog of *Defa2* cloned from C3H in previous research.[42] Seven published CRS peptides,[48] as well as three additional genes (*Crs4c5b*, *Crs4c7*, and *DefaL1*) and two new pseudogenes, were assembled. We realigned WGS Illumina reads from both C3H/HeJ and DBA/2J to their respective *de novo* assembly. Compared to the GRCm39 reference genome, SNPs or indels are reduced by over 99%, and the standard deviation in aligning depth is reduced by 75% in our *de novo* assemblies (Figure S6). Other previously cloned genes (e.g., *Defa8*, *Defa9*, *Crs4c1b*, etc.) cannot be found in our sequenced strains C3H/HeJ or DBA/2J.

The killer cell lectin-like receptor locus (*Klra*; also called the *Ly49* locus) in mice is another important gene family with significant numbers of non-reference genes, such as *Ly49L* (*Klra12*) in BALB or *Ly49T* (*Klra20*) in 129-related strains.[49] We have fully assembled five different haplotypes of the mouse *Ly49* (*Klra*) locus from 12 laboratory mouse strains (Figure S7).

### VNTRs in mouse protein-coding genes

Variable number tandem repeats (VNTRs) are composed of 10–100 bp tandem repetitive DNA with variable copy numbers and hyper-polymorphic sequences due to polymerase slippage during DNA replication.[50] VNTRs are also called "hidden polymorphisms" because their sequences are difficult to assemble using short-read-based approaches.[51] For example, SNP signals of indels in VNTRs are easily masked by similar sequences in the region, which results in fluctuations in coverage and a dozen heterozygous SNPs (Figure 3C). VNTRs, especially in protein-coding regions, are strongly associated with a wide spectrum of complex traits and diseases in humans.[52] Although several tools have been developed to discover VNTRs in genomes,[53,54] our understanding of the cause, prevalence, and function of VNTRs remains limited. Recently, we reported that the 6th exon of the mouse gene *Ifi207* encodes 11–25 copies of an exact 42 bp repeat, which have accumulated extremely high diversity (15% at the nucleotide level and 29% at the amino acid level) within a few million years.[55] The repeat regions of *Ifi207* are typical VNTRs, show essential roles for protein function, and are probably a result of host-pathogen coevolution.[55]

To discover VNTRs in other protein-coding genes at a genome level, we developed an AI-based tool for VNTR identification in protein-coding genes of *de novo* assemblies, with object detection based on computer vision on dot plot graphs between gene alleles (see STAR Methods). In a training and validation set consisting of 262 images (80% training and 20% validation), precision and recall were found to be 93.7% and 83.3%, respectively. VNTR polymorphisms were detected in 428 genes, around 3% of the total protein-coding genes in mice. Several gene families are enriched with VNTRs, for example, *Krab-Zfp* members, keratins

---

**Figure 2. Using the pangenome to interrogate the haplotype complexity of the H2 and α-defensin loci**
(A) Genome structure of H2 haplotypes from 12 classical laboratory mouse strains. A zoom-in of H2-K/A/B/E and H2-D/L/Q is shown at the top and bottom. To fit the current haplotype definitions of mouse H2, 8 haplotypes have a hierarchy from *b* to *z* (see STAR Methods).
(B) Maximum likelihood phylogeny of H2-K/D/Q/L alleles (bootstrap values are shown when >90). The color code is the same as (A).
(C) α-Defensin locus from GRCm39, DBA/2J, and C3H/HeJ beginning with gene *Crs4c6* and ending at *Defa-ps11* (left). The gene order and strand are indicated by black arrows. The maximum likelihood phylogenetic tree shows α-defensin homologs from GRCm39 and C3H/HeJ. Bootstrap values are shown when >90, and black arrowheads indicate *Defa2* in GRCm39 and C3H/HeJ.

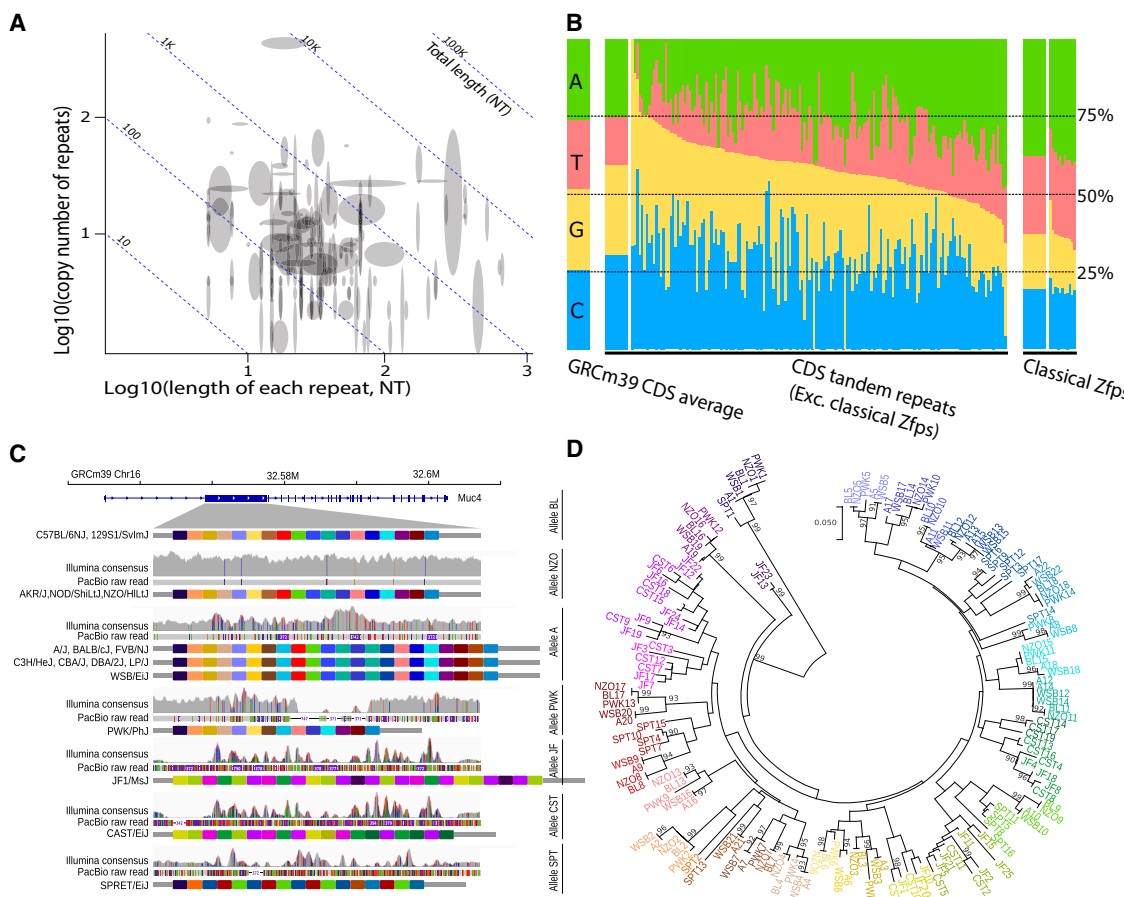

**Figure 3. VNTRs in mouse protein-coding gene regions**

(A) Statistics of copy number and length of each repeat in 252 genes. The height and width of the blocks indicate variations in copy number and repeat length. Homologous genes in the same family are merged into the same block and shown as a single gene.

(B) GC content of the VNTRs. Left, GRCm39 CDS regions average nucleotide content (GC ratio at 51.6%). Middle, nucleotide content in VNTRs in the CDS, without mouse Zfp members. The wider bar indicates the average value. Right, the GC ratio in "classical" 84 bp repeats in mouse *Zfp*.

(C and D) Tandem repeat diversity in mouse *Muc4* among 17 mouse strains. Each color block indicates a 278 bp repeat in *Muc4*. The colors are defined in (D), which are based on the phylogeny of the sequence. Closer colors indicate similar sequences.

(*Krt* members), keratin-related proteins (*Krtap* and *Tspear* family), and mucins (*Muc* family). Many of the genes have functions related to pathogen or cancer immunity (e.g., *Ubc*, *Ticam1*, *Stat2*, *Sbsn*, *Pierce1*, *Kmt2d*, *Mnda*, and *Ifi27L2b*), skin barrier (*Krt* members, *Muc* members, *Flg*, *Flg2*, *Hrnr*, *Rptn*, *Ivl*, *Kprp*, *Eppk1*, and *Tchh*), and sperm development (*Zan*, *Txndc2*, *Tsga8*, *Tro*, *Tex44*, *Semg1*, *Qrich2*, *Speer4*, *Mageb4*, *Fscb*, etc.).

The length of VNTR units ranges from a few bp up to 1 Kbp, with more than half of the protein-coding VNTRs having a repeat unit length from 10 to 100 bp (Figure 3A). The longest repeat unit is found in *Eppk1* (984–990 bp). For copy numbers, more than half of the VNTRs range from 5 to 20 copies, with a maximum of ~680 copies (*Muc17* in JF1/MsJ). Most VNTR sequences (251 genes with a repeat length > 3 bp) have a significantly higher GC ratio ($p < 0.01$, two-tailed $t$ test) than the whole open reading frame (ORF) average, while "classical" *Zfp* members with 84 bp repeats have a significantly lower GC ratio ($p < 0.01$; Figure 3B; Table S7). The exact sequence of long VNTRs has been extremely

challenging to determine; we identified different copy numbers of repeats between C57BL/6NJ and GRCm39 in the sequence of *Ubc*, *Ahnak2*, *Flg*, *Gm5154*, and *B230307C23Rik* and then confirmed them as GRCm39 assembly mistakes by cross-checking them with the BL6 T2T assemblies.[29,30] The *Flg* gene has four different sequences in GRCm39, C57BL/6NJ, and the two recent mouse T2T assemblies,[29,30] indicating mistakes in at least three assemblies or accelerated evolution of the coding sequence in a small number of generations.

VNTRs are reported to be hyper-mutable through motif copy-number changes due to polymerase slippage during DNA replication,[50] so shorter repeats with higher copy numbers should accumulate more mutations. In our genome-wide scan, we found variable VNTR examples with different evolutionary profiles. Some VNTRs have conserved shorter repeats (e.g., *Fam186a*, 12–25 copies of 57 bp repeats, Tajima's $\pi = 7.99 \times 10^{-2}$ between repeat units) and long repeat units with high diversity (e.g., *Muc4* with 14–27 copies of 369–372 bp repeats, $\pi = 0.107$). Also, we

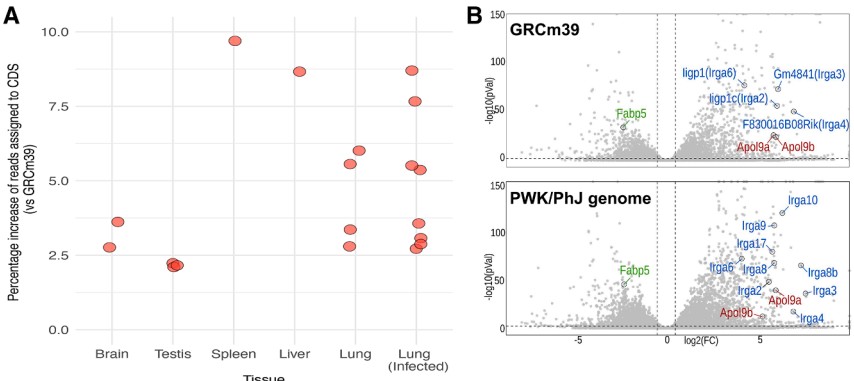

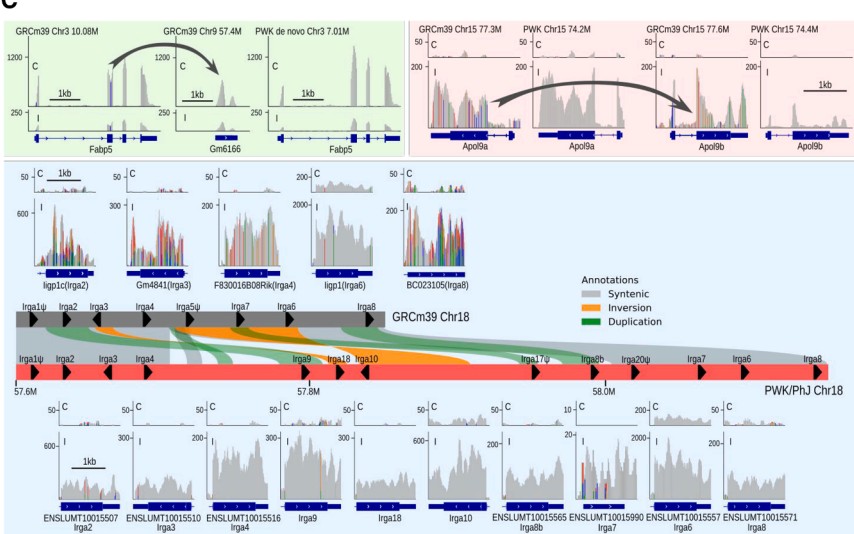

**Figure 4. Strain assemblies provide more accurate gene expression quantification**

(A) Increase in proportion of RNA-seq reads mapped to protein-coding sequence (CDS) for the PWK/PhJ strain when using the PWK/PhJ reference genome compared to GRCm39. RNA-seq data from normal and infected tissues are shown (see STAR Methods).

(B) Gene expression volcano plots from differential gene expression analysis of PWK/PhJ lung SARS-MA15 infection vs. control.[58] The top image uses the GRCm39 reference genome, and the bottom image uses the PWK/PhJ reference genome. Dashed lines in the volcano plot indicate adjusted $p < 0.01$ and log2 (fold change) (log2(FC)) > 0.5. The labeled genes are explored in (C), in images corresponding to the label color.

(C) Examples visualized by IGV are shown: *Fabp5* (green), where there is a change of pseudogene copy number between GRCm39 and PWK/PhJ; *Apol9a* and *Apol9b* (red), where there is a misalignment of reads caused by high-density SNPs in the coding sequence; and *Irga* locus (blue), in which there are several true novel genes in the PWK/PhJ genome. Arrows indicate raw reads aligned onto incorrect regions.

found highly conserved repeating sequences (e.g., *Ubc*, 8–23 copies of 228 bp repeats, Tajima's $\pi = 8.95 \times 10^{-3}$). In addition, we have observed similar tandem repeat units between different subspecies of mice, e.g., *Muc4* in WSB/EiJ (*M. m. domesticus*) shares similar units with PWK/PhJ (*M. m. musculus*) and JF1/MsJ (*M. m. molossinus*) shares sequences with CAST/EiJ (*M. m. castaneus*; Figure 3D). These results strongly indicate a long-lasting balancing selection rather than random mutation.

### *De novo* genome assemblies improve gene expression quantification

Gene expression quantification using RNA-seq is a fundamental experimental assay in biomedical research.[56] Until now, most gene expression studies of different mouse strains have used the BL6 reference genome despite millions of known genetic variations between strains.[14,15] Previous studies have shown that using a strain consensus genome produced by inserting SNPs and indels into GRCm39 improves RNA-seq quantification.[57] We aligned RNA-seq from several tissues isolated from PWK/PhJ to both the GRCm39 reference and the PWK/PhJ reference genome (see STAR Methods). The fraction of RNA-seq reads that align to CDS is 2%–10% higher across different tissues when aligned to the PWK/PhJ reference, compared to GRCm39 (Figure 4A). We have also compared the gene expres-

sion levels from lung tissue infected with MA15 SARS virus and uninfected control[58] using both reference genomes. We identified 359 genes (2.1% of 16,222 genes with clear 1-to-1 paralogs) that are either differentially expressed in the opposite direction or change to a differential expression state (or vice versa) (Figures 4B and S8). We identified three distinct scenarios where these differences occur. In scenario 1, an additional processed pseudogene copy in the GRCm39 reference attracts reads from the original gene. For example, in GRCm39, pseudogene *Gm6166* reduced the observed expression of *Fabp5* (Figures 4B and 4C, green). In scenario 2, sequence similarity between gene family members and SNP differences between strains cause the misalignment of reads between gene homologs. For example, multiple SNPs in the first exon of PWK/PhJ *Apol9a* make it more similar to BL6 *Apol9b* than to *Apol9a*, so both genes have the incorrect expression levels when GRCm39 is used as the reference genome (Figures 4B and 4C, red). In scenario 3, strain-specific members of the *Irga* gene family in PWK/PhJ are expressesed during acute infection. Novel genes such as *Irga9*, *Irga10*, and *Irga18* do not exist in BL6, so all the raw reads are misleadingly aligned to *Irga2*, *Irga3*, and *Irga8* (Figures 4B and 4C, blue).

### Structural variation in the mammalian apolipoprotein L gene cassette underlies tuberculosis resistance

In a recent study, a cohort of 52 genetically diverse CC mouse strains, in conjunction with a transposon mutant library of *Mycobacterium tuberculosis* (*Mtb*), the causal agent of tuberculosis, was used to create a resource for associating bacterial

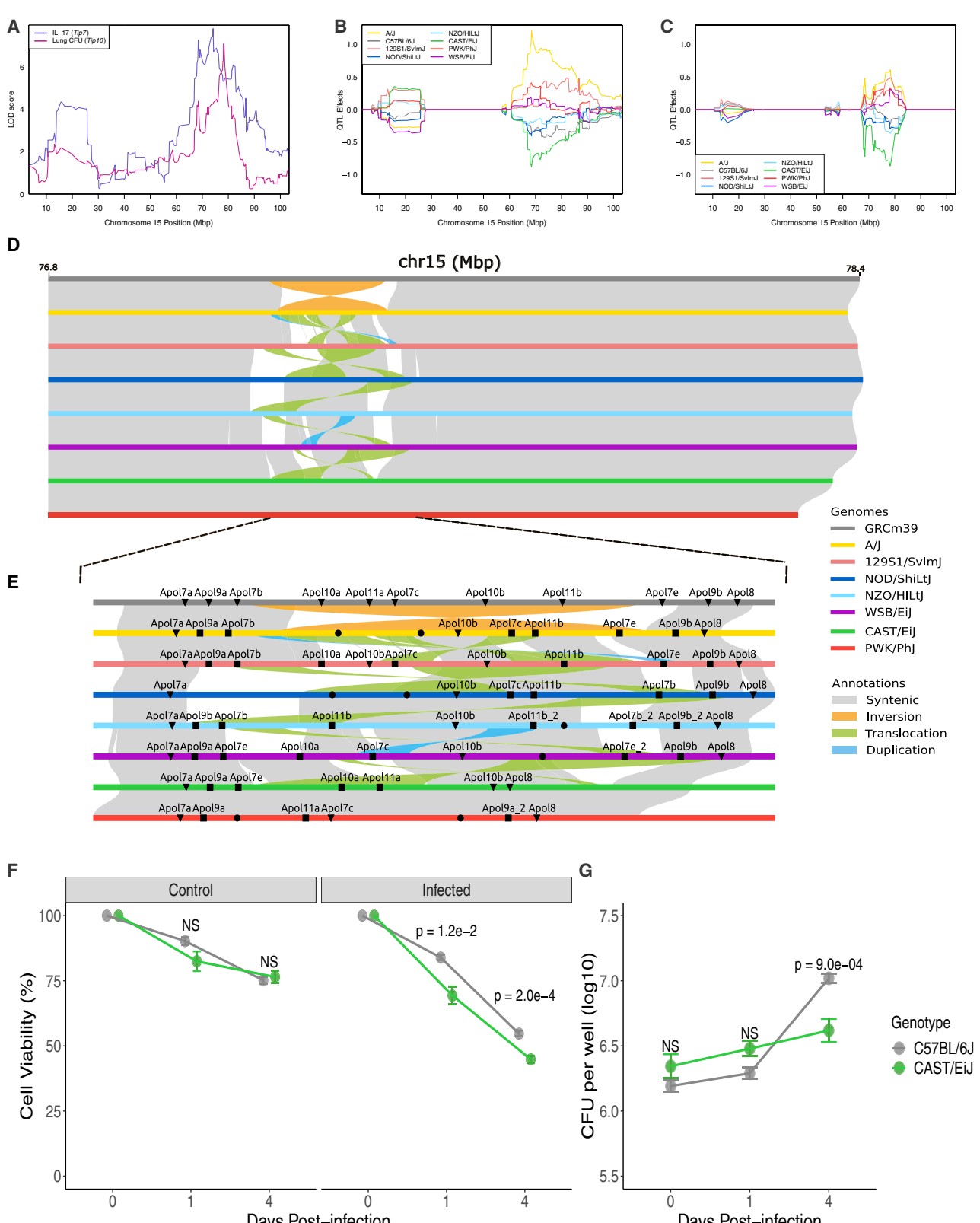

*(legend on next page)*

genetics with host genetics and immunity.[59] CC strains vary dramatically in their susceptibility to infection and produce qualitatively distinct immune states. Two quantitative trait loci (QTLs), representative of pulmonary interleukin (IL)-17 levels and bacterial burden at 4 weeks post-infection, were mapped to a chr15 locus (Figure 5A). For both phenotypes, inheritance of the CAST/EiJ haplotype at this locus was predictive of lower values by best linear unbiased prediction (BLUP) analysis (Figures 5B and 5C). To measure substantial changes in genomic sequence from the GRCm39 reference genome from BL6, chr15 was scanned for structural variants and was found to be highly polymorphic (Figure 5D). Across the CC/DO founders, a non-syntenic region of approximately 0.4 Mbp in length was identified on chr15, containing an inversion present in BL6 and 129S1/SvImJ and not found in the remaining seven founder lines (Figure 5E). Leveraging novel RNA-seq-based gene annotations, each of the non-reference founders was found to harbor 6–8 de novo protein-coding genes within this non-syntenic region (Figure 5E). Evidence suggests that genes within this region in CAST/EiJ mice meaningfully promote resistance to pathogenic infection. Previously, the *Apol* genes found within the divergent chr15 locus were hypothesized to promote host resistance to pathogenic infection through directly targeting bacteria in the host cell cytosol and cell death mechanisms.[60,61] In a bone-marrow-derived macrophage (BMDM) infection model, CAST/EiJ macrophages exhibited earlier cell death than BL6 macrophages following infection (Figure 5F), which corresponded to enhanced *Mtb* restriction by 4 days post-infection (Figure 5G). Previous transcriptomic studies of infected inbred BL6 and C3HeB/FeJ mice[62] and outbred DO mice[63] reveal that the majority of the *Apol* gene family on chr15 is significantly upregulated in mouse lungs following *Mtb* infection. We report extensive variation between protein-coding sequences of predicted *Apol* genes found within this region among the CC/DO founder lines, suggesting that this coding variation may differentially impact the immunogenetic response to *Mtb* infection between genetically divergent mouse lines. Collectively, this evidence supports previous hypotheses suggesting that host-driven cell death pathways, regulated by the *Apol* gene cassette on chr15, promote resistance to pathogenic infection.

## DISCUSSION

Mice are widely used in human disease research. By relying on a single reference genome, researchers are blind to genes and structural variants present in other mouse strains, potentially leading to incomplete or biased results. The existence of a single reference genome inherently shifted scientists' preferences when choosing an experimental mouse strain, which further reinforced this bias. Compared to previous *de novo* mouse strain genomes and variation catalogs,[14,15,18] these long-read genome assemblies show significant improvement in the most complex regions, such as MHC, defensin, T cell receptor (TCR), and *Ly49* loci. Hundreds of non-reference genes discovered and published in historical literature but absent from the BL6-based reference have now been located in these new genomes, including *Defa1*, *Raet1a*, and *Klra19-28* (*Ly49S*, *Ly49T*, *Ly49U*, *Ly49V*, etc.). These pangenomic data contextualize previously published results and facilitate future research in immunity, neuron development, behavior, sensory functions, and reproduction.

VNTRs are a relatively unexplored area of mouse genome variation, as they are not easily detected with short-read-based methods. Until now, there have been no genome-wide reports of VNTRs in the mouse genome. We have developed an AI-based method to perform a genome-wide scan of VNTRs in the CDS of all annotated genes among 17 mouse strains. Surprisingly, a large number of genes (426, around 3% of protein-coding genes in mice) have VNTRs in their coding region, most of which have not been reported before. We found VNTR-related sequence errors in the CDS of 5 genes in GRCm39. Many of the newly discovered VNTR genes play important roles in human disease, e.g., *STAT2* in antiviral immunity,[64] *HRNR* in atopic dermatitis skin lesions[65] and hepatocellular carcinoma,[66] *SBSN* in tumor progression,[67] *KMT2D* in Kabuki syndrome,[68] *FLG2* in peeling skin syndrome,[69] *KRTAPs* in ectodermal dysplasia,[70] *MAGEB4* in azoospermia,[71] and *RAD18* in Alzheimer's disease.[72] Currently, our method is not sensitive to short VNTRs because they are not readily visible in the dot plot. Nevertheless, these mutations can be detected accurately with short-read-based methods. Previous research reports polymerase slippage during DNA replication as the primary mechanism of VNTRs.[50] However, our data indicate that VNTRs in protein-coding genes are not caused by a single mechanism. Like other regions in the mouse genome, purifying selection and balancing selection can be found in genes with significant VNTRs, and tandem repeats might be strictly conserved in such genes. In short, VNTRs might be a genome phenomenon caused by different reasons and mechanisms. Polymerase slippage only happens in some cases of VNTRs, which accelerates the accumulation of mutations.

**Figure 5. Extensive structural variation on chr15 is associated with CAST/EiJ resistance to *Mtb* infection, mediated through host cell death**
(A) Association of lung IL-17 levels (*Tip7*) and lung *Mtb* burden (*Tip10*) across chr15 in a collaborative cross (CC) infection cohort.
(B and C) Best linear unbiased predictions (BLUPs) of allele effects underlying (B) *Tip7* and (C) *Tip10* in the CC infection screen. QTL effect values correspond to log10-transformed phenotype values.
(D) Structural variation between the eight CC/DO panel founder strains within the QTL causal interval, identified through the lift over of flanking sequences.
(E) Annotated and previously unannotated mouse genes in CC/DO founder genomes. Triangles indicate genes that were directly annotated and named in the current assembly annotation, squares indicate genes that had to be renamed according to synteny with GRCm39 genes, and circles indicate *de-novo*-annotated genes that could not be named.
(F) Cell viability of C57BL/6J and CAST/EiJ BMDMs infected with *Mtb* (MOI 10), determined via enzyme lactate dehydrogenase (LDH) release assay.
(G) Intracellular colony-forming units (CFUs) per well were subsequently assessed via dilution plating of lysed BMDMs. Measurements were collected 1 and 4 days post-infection (representative of 2 experiments; 4 technical replicates per experiment).
Boxes in (F) and (G) are colored by host genotype. Hypothesis testing was performed using ANOVA and Tukey's post hoc test.

Our mouse pangenome will also catalyze research using non-BL6 strains in areas such as immunity, sensory, and neuron research by providing essential genomic resources. For example, our strain-specific genomes and annotations significantly improve the accuracy of RNA-seq and single-cell RNA-seq analysis. On average, we observed a 5% improvement in short-read mapping, with more than 2.5% of coding genes showing significant differences in expression level compared to GRCm39. In addition, strain-specific genomes also contain novel non-reference genes, which will be particularly useful for fine mapping in highly complex and immunologically impactful loci (e.g., Figure 5). This mouse pangenome consists of 17 high-quality reference genomes, but further iterations will be reported in the coming years as the number of mouse strain genomes sequenced to T2T quality increases.[29,30]

## Limitations of the study

While this study represents a significant step toward complete genomes for the most commonly used mouse strains, the most significant limitation is that these genomes are not T2T quality and therefore lack sequences for the most complex regions, such as the rDNAs, telomeres, and centromeres, and the Y chromosome. There are other key questions in mouse genetics, such as the existence and extent of chromosomal heterogeneity in inbred mouse strains, that we believe can only be answered with T2T-quality genomes.

## RESOURCE AVAILABILITY

### Lead contact

Requests for further information and resources should be directed to and will be fulfilled by the lead contact, Thomas M. Keane (tk2@ebi.ac.uk).

### Materials availability

This study did not generate new unique reagents.

### Data and code availability

- The genome sequencing reads and the genome assemblies generated in this study have been deposited at the European Nucleotide Archive (ENA: PRJEB47108). The genomes and annotation are available from the Ensembl genome browser (https://projects.ensembl.org/mouse_genomes/). RNA-seq accessions are given in Table S4.
- No other custom code/software was used for data analysis in the study. The publicly available software and algorithms used in the present study are listed in the key resources table.

## ACKNOWLEDGMENTS

Funding for this work was provided by the UK Medical Research Council grant (MR/R017565/1) and the European Molecular Biology Laboratory (T.M.K. and M.H.), the National Institute on Drug Abuse (U01DA059695) (P.B.C. and J.F.), the Fulbright Association and the Coordenação de Aperfeiçoamento de Pessoal de Nível Superior (88881.625374/2021-01) (M.T.P.G.), Wellcome (220540/Z/20/A) (D.J.A.), Wellcome (WT222155/Z/20/Z) and the European Molecular Biology Laboratory (L.H. and F.J.M.), Wellcome (219475/Z/19/Z and 092096/Z/10/Z) (E.A.M.), and the NIH Director's New Innovator Award (AI183152) (C.M.S. and R.K.M.).

## AUTHOR CONTRIBUTIONS

This study was conceived by T.M.K., J.L., and C.M.S. Sample provision and genome sequencing were carried out by E.A., P.B.C., A.M.C., T.D.D., J.F., E.H., E.A.M., I.R., N.M., N.B.R., A.S., L.R., and D.J.A. The data analysis was performed by M.H., J.U.L., X.F.Y., R.K.M., E.G., M.T.P.G., A.G., K.H., D.S., R.Q., J.M.D.W., T.W., and D.J.Z. Genome annotation was carried out by L.H., F.J.M., S.S., and F.F.T. T.M.K., J.L., C.M.S., and M.H. wrote the manuscript. All authors have read and approved this manuscript.

## DECLARATION OF INTERESTS

The authors declare no competing interests.

## STAR★METHODS

Detailed methods are provided in the online version of this paper and include the following:

- KEY RESOURCES TABLE
- EXPERIMENTAL MODEL AND STUDY PARTICIPANT DETAILS
- METHOD DETAILS
  - Hi-C sequencing
  - Mouse testes RNA-Seq
  - De novo assembly
  - Assembly QC
  - Gene annotation
  - Assembly graphs
  - Synteny and phylogeny
  - MHC haplotypes
  - Non-reference gene rediscovery
  - RNA-seq quantification
  - Variable number tandem repeats (VNTR) in coding regions

## SUPPLEMENTAL INFORMATION

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

# STAR★METHODS

## KEY RESOURCES TABLE

| REAGENT or RESOURCE | SOURCE | IDENTIFIER |
| --- | --- | --- |
| **Deposited data** | | |
| Arima Hi-C kit | Arima | A510008 |
| **Experimental models: Organisms/strains** | | |
| C57BL/6NJ | Jackson Laboratory | 005304 |
| FVB/NJ | Jackson Laboratory | 001800 |
| A/J | Jackson Laboratory | 000646 |
| AKR/J | Jackson Laboratory | 000648 |
| BALB/cJ | Jackson Laboratory | 000651 |
| C3H/HeJ | Jackson Laboratory | 000659 |
| CBA/J | Jackson Laboratory | 000656 |
| CAST/EiJ | Jackson Laboratory | 000928 |
| DBA/2J | Jackson Laboratory | 000671 |
| LP/J | Jackson Laboratory | 000676 |
| NOD/ShiLtJ | Jackson Laboratory | 001976 |
| NZO/HlLtJ | Jackson Laboratory | 002105 |
| PWK/PhJ | Jackson Laboratory | 004660 |
| SPRET/EiJ | Jackson Laboratory | 001146 |
| WSB/EiJ | Jackson Laboratory | 001145 |
| 129S1/SvlmJ | Jackson Laboratory | 002448 |
| JF1/MsJ | Jackson Laboratory | 003720 |
| **Software and algorithms** | | |
| samtools and bcftools | www.htslib.org | v1.4 |
| Falcon | https://github.com/PacificBiosciences/FALCON | v0.3.0 |
| Hifiasm | https://github.com/chhylp123/hifiasm | v0.15.1 |
| purge_dups | https://github.com/dfguan/purge_dups | v1.0.0 |
| minimap2 | https://github.com/lh3/minimap2 | v2.17 |
| BESST | https://github.com/ksahlin/BESST | v2.2.4 |
| Bionano Solve | https://bionano.com/software-downloads/ | v3.2.2 |
| SALSA | https://github.com/marbl/SALSA | v2.2 |
| Arima Genomics pipeline | https://github.com/ArimaGenomics | v1.0 |
| Arrow | https://github.com/PacificBiosciences | v1.0.0 |
| Pilon | https://github.com/PacificBiosciences | v1.23 |
| TGS-GapCloser | https://github.com/BGI-Qingdao/TGS-GapCloser | v1.1.1 |
| RagTag | https://github.com/malonge/RagTag | v1.0.0 |
| BWA-MEM2 | https://github.com/lh3/bwa | v2.2.1 |
| PretextMap | https://github.com/sanger-tol | v0.1.9 |
| PretextView | https://github.com/sanger-tol | v0.2.5 |
| QUAST | https://github.com/ablab/quast | v5.0.2 |
| BUSCO | https://busco.ezlab.org/ | v3.1.0 |
| RepeatMasker | https://www.repeatmasker.org/ | v4.1.2-p1 |
| Mercury | https://github.com/marbl/merqury | v1.3 |
| minigraph | https://github.com/lh3/minigraph | v0.19 |
| gfatools | https://github.com/lh3/gfatools | v0.4 |
| Sushi R package | https://github.com/PhanstielLab/Sushi | v1.32.0 |

*(Continued on next page)*

 CellPress

**Cell Genomics**
Article

***Continued***

| REAGENT or RESOURCE | SOURCE | IDENTIFIER |
|---|---|---|
| SyRI | https://schneebergerlab.github.io/syri/ | v1.6.3 |
| plotsr | https://github.com/schneebergerlab/plotsr | v1.1.1 |
| gffread | https://github.com/gpertea/gffread | v0.12.7 |
| miniprot | https://github.com/lh3/miniprot | v0.12 |
| pangene | https://github.com/lh3/pangene | v1.1 |
| bandage | https://github.com/rrwick/Bandage | v0.8.1 |
| MEGA | https://www.megasoftware.net/ | v11 |
| STAR | https://github.com/alexdobin/STAR | v2.7.10 |
| DEseq2 | https://bioconductor.org/packages/release/bioc/html/DESeq2.html | v1.34.0 |
| yolov10 | https://github.com/THU-MIG/yolov10 | v10 |
| IGV | https://igv.org/ | v2.18 |

## EXPERIMENTAL MODEL AND STUDY PARTICIPANT DETAILS

Flash-frozen tissue was obtained from the Jackson Laboratory from female mice. DNA was extracted from kidney tissue using the Qiagen MagAttract HMW DNA Kit according to the manufacturer's protocol. For DBA/2J, LP/J, NZO/HILtJ, and JF1/MsJ, the Pacbio SEQUEL platform was used to generate HiFi reads according to the manufacturer's protocols. For the rest of the strains, the Pacbio SEQUEL platform was used to generate high-depth CLR reads per strain according to the manufacturer's protocols.

10 Kb Illumina Nextera libraries were prepared according to the manufacturer's instructions (Illumina Nextera Sample Preparation guide) with the addition of a size selection step on the BluePippin (Sage Science; Beverly, MA, USA), and sequenced on the Illumina HiSeq X platform. Pacbio sequencing depths per strain are given in Table S1.

Every sequencing run was genotype checked against the mouse Hapmap SNP calls[73] using the Samtools/Bcftools v1.14 'gtcheck' command.[74]

```
bcftools mpileup -f reference.fasta '{strain}'.bam | bcftools call -c -Oz -o '{strain}'.vcf.gz && bcftools index -t '{strain}'.vcf.gz
bcftools gtcheck -G 1 '{strain}'.vcf.gz -g refrence_genotypes.vcf.gz > '{strain}'genotypes
```

## METHOD DETAILS

### Hi-C sequencing

For strains 129S1/SvlmJ, AKR/J, C3H/H3J, C57BL/6NJ, CBA/J, JF1/MsJ, LP/J, NOD/ShiLtJ, NZO/HILtJ, and SPRET/EiJ, Flash frozen kidney tissue was obtained from the Jackson Laboratory from female mice. Hi-C sequencing libraries were prepared using the Arima proximity ligation method following the manufacturer's protocols for tissue preparation. Libraries were sequenced on the Illumina NovoSeq platform.

For strains A/J, BALB/cJ, FVB/J, DBA/2J, WSB/EiJ, PWK/PhJ, and CAST/EiJ: Male mice from seven inbred strains were purchased from Jackson Laboratories at 8 weeks of age and transferred to UCLA where they were kept for at least 7 days before tissue extraction. Adult male animals (Jackson Laboratories) were euthanized at 10–16 weeks old in an isoflurane chamber and decapitated. Hippocampal dissections, the brain was removed and the ventral region of the hippocampus was microdissected, snap frozen in dry ice, and stored at −80 until processing. Amygdala dissections, the brain was removed and coronal brain slices containing amygdala tissue were generated on a 1mm brain matrix (World Precision Instruments). Amygdala tissue was microdissected from these slices under a dissecting scope in cold PBS, snap frozen in dry ice, and stored at −80 until processing. Tissue from ~2 to 3 animals were combined into a single tube and considered a replicate. We generated Hi-C libraries using the Arima Genomics workflow. Hi-C libraries were generated using the Arima-HiC kit (A510008) with library preparation using Swift Biosciences Accel-NGS 2S Plus DNA Library and Indexing Kits (#21024, #26396). All steps were carried out according to the manufacturer's protocols for animal tissue (documents A160132 v01, A160140 v00), including quality control for libraries. Libraries were sequenced on a Novaseq 6000 with 150bp paired-end reads.

## Article

CellPress

### Mouse testes RNA-Seq

Testes from P20.5 (20.5 dpp) male mice were used for all 16 strains except JF1/MsJ. Three biological replicates of pairs of testes from individual mice were dissected, flash-frozen and shipped to the UK on dry ice for further processing. Each biological replicate was derived from independent litters. RNA extraction, library prep and initial sequencing with preliminary QC analysis were performed on a subset of samples prior to application on all remnant samples. Single P20.5 testis were first homogenised in Qiazol. The miRNeasy-mini kit (Qiagen) was then used to extract >200nt RNA fraction according to the manufacturer's instructions. Eluted RNA fractions were measured using Qubit RNA Broad Range assay (Invitrogen) and stored at −80°C. Tapestation RNA Screentape (Agilent) QC (RIN >9.0) was subsequently performed by the Wellcome Sanger Institute core sequencing facility, and ribosomal RNA-depleted RiboZero libraries were made from 1 μg of each of the RNA fractions using the TruSeq Stranded Total RNA kit (Illumina). Multiplexed libraries were sequenced on the Illumina HiSeq2000 instrument to generate Paired-End (PE) 75 bp reads.

### De novo assembly

Initial contigs were assembled using PacBio's Falcon software and Hifiasm (v0.15.1) for CLR and HiFi PacBio reads respectively. CLR reads were purged using purgedups (v1.0.0). Scaffolding was incrementally aided by different types of long-range fragments. Firstly, mate-pairs 10kb reads were aligned to the contigs assemblies using minimap2 (v2.17) and scaffolded using BESST (v2.2.4), BioNano optical maps were then used to further scaffold using Solve (v3.2.2), Finally, SALSA (v2.2) was used to produce the highest level scaffolds using Hi-C reads that underwent preprocessing using The ArimaGenomics mapping pipeline (https://github.com/ArimaGenomics). CLR assemblies were subject to two rounds of polishing to correct for base errors. Initially, CLR subreads were realigned to contigs using minimap2 wrapper pbmm2 and PacBio's GCpp Arrow algorithm (v1.0.0) was used to polish the assemblies. Secondly, Illumina paired-end reads (https://www.mousegenomes.org/mouse-strains-sequenced/) were aligned to the scaffolds using minimap2 (v2.17) and polished using Pilon (v1.23). Furthermore, gap filling was performed on CLR-based assemblies using TGS-GapCloser (v1.1.1). Assembled scaffolds were then arranged into chromosomes with RagTag (v1.0.0) and using GRCm39 assembly as a guide. Finally, Hi-C reads were realigned to the chromosome level assemblies using BWA-MEM2 (v2.2.1) and contact maps were generated with PretextMap (v0.1.9). The Hi-C contact maps were used to further curate the assemblies ensuring that all sequences are placed in the correct order and orientation to chromosomes using PretextView (v0.2.5) and final curated assemblies generated with rapid_join.pl (https://gitlab.com/wtsi-grit/rapid-curation).

### Assembly QC

Assembly metrics were collected using QUAST (v5.0.2) and BUSCO (v3.1.0, metazoa_odb10) was used to assess the completeness of the gene contents. RepeatMasker (v4.1.2-p1) was used to annotate and mask the interspersed repeat sequences across the assemblies. Finally, Mercury (v1.3) was used to assess base accuracy of the assemblies through calculating average QV scores using Illumina short reads (k = 21).

Raw Illumina reads were downloaded from the Mouse Genomes Project website (https://www.mousegenomes.org/mouse-strains-sequenced/), BL6 reads from EVE (accession Genbank: LXEJ00000000), and mhaESC T2T from NCBI BioProject PRJNA1097000. Reads were aligned to the strain reference genomes with BWA (v0.7.17). Downstream SNP, indel, and ultralong insertion calling are finished by samtools (v1.18) and bcftools (v1.18) with the following commands.

#### Read mapping

```
bwa mem -t 12 -CH Grap <Ref.fa> <reads_1.fastq> <reads_2.fastq> | samtools sort -m 8G -@ 4 -O bam -l 0 -T ./ - | samtools view -T <Ref.fa> -C -o out.cram -
```

#### Variant calling

```
bcftools mpileup -Ou -g 10 -a FORMAT/DP,FORMAT/AD,FORMAT/ADF,FORMAT/ADR,FORMAT/SP,INFO/AD -E -Q 0 -pm3 -F0.25 -d 500 -f
<Ref.fa> <out.cram> | bcftools call -mv -f GQ,GP -p 0.99 | bcftools norm --fasta-ref $2 --rm-dup all --multiallelics +indels
--strict-filter | bcftools filter -s SnpGap -m + --SnpGap 2 -s IndelGap -m + --IndelGap 3 | bcftools filter -s LowQual -m + -e
'QUAL<20' | bcftools filter -s LowDP -m + -e 'INFO/DP<5' | bcftools filter -s HiDP -m + -e ''INFO/DP>120'' | bcftools filter
-s MinDP4 -m + -e 'INFO/DP4[2]+INFO/DP4[4]<5' | bcftools filter -s Het -m + -e 'GT!=''1/1''' | bcftools filter -s RefN -m +
-e 'REF~''N''' -Oz -o <out.vcf.gz>
```

**CellPress**

**Cell Genomics**
Article

### SNP counting

```
zcat <out.vcf.gz> | grep -v 'ˆ#' | grep -v 'INDEL' | grep PASS | wc -l
```

### Indel counting

```
zcat <out.vcf.gz> | grep -v 'ˆ#' | grep $'\tINDEL' | grep PASS | wc -l
```

### Pairs with >5K insertion

```
samtools view -h -T <Ref.fa> | awk 'substr($0,1,1)==''@'' || ($9>=5000) || ($9<=-5000)' | wc -l
```

The alignment and variant calling results are shown in the Table S3. The only exception is the SNP count of the strain WSB/EiJ. Bcftools reported 20656 SNPs with Q > 20, which is five times higher than any other strain. A double-check of aligned raw data has confirmed multiple low-quality regions in the *de novo* assembly caused by low PacBio raw read coverage. These regions are reported as "low quality regions" in sum 2574838 base pairs, containing 16856 SNPs. The list of affected regions is available as BED format in Data S2.

### Gene annotation

Gene sets for the mouse strain assemblies were generated using a combination of the Ensembl mapping pipeline - originally developed for human pangenome annotation, and the Ensembl vertebrate automated annotation system. This integrated approach leverages strain-specific short-read RNA-seq and long-read transcriptomic data to produce high-confidence annotations.

A subset of GENCODE M30 [https://www.gencodegenes.org/mouse/release_M30] genes and transcripts (GENCODE M30) was annotated onto each haploid assembly using the Ensembl mapping pipeline. This subset excludes readthrough genes as well as genes located on patches or alternate haplotypes. For each gene, anchor sequences derived from the flanking regions were used to locate the most likely corresponding region in the target genome. Pairwise alignments between the reference and target sequences enabled the projection of exon coordinates and other gene features onto the target assembly. To identify recent duplications and potential collapsed paralogues, canonical transcripts were aligned genome-wide, and novel mappings that did not overlap existing annotations were flagged for further consideration. For additional details on the annotation process, refer to the Methods section, Ensembl Mapping Pipeline for Assembly Annotation, in the publication, A draft human pangenome ref.[75]

This approach was supplemented with methods from the Ensembl vertebrate annotation pipeline. The genome was masked and repeats annotated using RepeatMasker (version 4.0.5; parameters: -nolow -engine "RMBlast", Rodentia Repbase library) (Smit, AFA, Hubley, R., & Green, P., 2013–2015), Dustmasker,[76] and Tandem Repeat Finder (TRF).[77] Genome annotation was driven by alignment of publicly available transcriptomic data. Strain-specific RNA-seq data were downloaded from the European Nucleotide Archive (ENA) and aligned to the genome using STAR.[78] Transcript models were then assembled using Scallop.[79] Long-read transcriptomic data were also obtained from ENA and aligned to the genome using Minimap2[80] with recommended settings for Iso-Seq data.

Protein-coding models were validated by aligning the longest open reading frame (ORF) against a mammalian SwissProt protein database using DIAMOND.[81] Translations of immunoglobulin gene segments from closely related species were downloaded from the IMGT database[82] and aligned to the genome using GenBlast. For GenBlast, the following thresholds were applied: minimum 80% coverage, 70% identity, and an e-value of $\leq$ 1e−1, with exon repair enabled. Up to 10 top-scoring transcript models per protein were retained.

Low-quality models were filtered out, and the remaining data were collapsed and consolidated into a final set of gene models and non-redundant associated transcripts. Where available, annotations derived from mapping the reference genome were prioritised. In cases of fragmentation or absence, transcriptomic evidence was used to fill gaps.

### Assembly graphs

All strain assemblies, in addition to GRCm39 were incorporated into a pangenome graph using minigraph (v0.19).[83] GRCm39 assembly was used as the backbone of the graph where SVs were reported relevant to the GRCm39 sequence and using its coordinates.

The strain coordinates of SVs were then extracted using gfatools (v0.4).[84] Bedgraph-like files were then extracted from minigraph output showing chromosome, start, end, and amount of divergence for each SV in each strain relative to GRCm39 (Data S3). Sushi R package (v1.32.0)[85] was then used to visualise the distribution of such diverse regions across strains.

### Synteny and phylogeny

Gbp and Apol regions were extracted from each assembly using samtools (v1.17),[74] and minimap2 (v2.17)[80] was used to align the regions between GRCm39 and wild-derived strains (Gbp) and the CC strains (Apol), incrementally. Python packages SyRI (v1.6.3)[86] and plotsr (v1.1.1)[87] were then used to identify and visualise synteny and structural rearrangements across the region.

For the GBP locus, gffread (v0.12.7)[88] was used to extract coding sequences of each of the genes within the region across all wild-derived strains and GRCm39. These genes were then aligned to each of the genomes (GRCm39, WSB/EiJ, CAST/EiJ, PWK/PhJ, JF1/MsJ, SPRET/EiJ) using miniprot (v0.12)[89] and pangene (v1.1)[90] was then used to create a pangene graph incorporating alignments to all strains. This graph was visualised using bandage (v0.8.1).[91] Gene names were updated according to phylogeny with GRCm39 genes (Data S4).

### MHC haplotypes

For each strain, a contig of DNA from the gene *Tapbp* to *Trim26* was extracted from the *de novo* assemblies, based on their relately conserved nucleotide sequences. Synteny between strains was based on dot plot results, generated by LBDot.[92] For haplotype analysis, assembly of *H2* haplotype *a, bc, k, d, q, g7,* and *z* was sliced into 100 Kb fragments and aligned onto the *H2* contig of all other haplotypes, with minimap2. With a hierarchy $b{\rightarrow}k{\rightarrow}d{\rightarrow}a{\rightarrow}q{\rightarrow}c{\rightarrow}g{\rightarrow}z$, we check the similarity of a haplotype to all other haplotypes with a higher hierarchy. If a continuous contig >10K in length has less than 3 SNPs (0.3‰ SNPs) to high-hierarchy-haplotypes, the region will be marked with the haplotype with the highest haplotype. For example, if a 10 Kb region in haplotype *z* has <3 SNPs against the synteny region to both haplotype *k* and haplotype *b*, this will be marked with haplotype *b* because *b* has the highest hierarchy.

The value 0.3% was chosen based on the evolutionary history of laboratory mice and mouse strains. We assume modern mouse strains have a ~400-year history since fancy mouse breeding, during this period, around 0.3% mutations may accumulate under neutral selection.

The phylogenetic tree of MHC alleles was inferred by using the Maximum Likelihood method and the Poisson correction model. The analyses were conducted in MEGA11.[93]

### Non-reference gene rediscovery

A list of all gene markers was downloaded from the MGI database, Entrez gene list (https://www.informatics.jax.org/downloads/reports/MGI_EntrezGene.rpt), and screened for gene markers not found in GRCm39 ("non-withdrawn" and without "genome coordinate"). Qualified gene markers were manually checked on both the MGI and the NCBI databases for previous publications, ESTs, or nucleotide sequences. Other gene makers are collected from the Celera mouse assembly[94] or publications on several gene families, including *Defa* members. Nucleotide sequences of genes from non-reference mouse strains were aligned to corresponding *de novo* assemblies. If the gene belonged to a strain without *de novo* assembly, or outbred strains, the sequences were aligned to all *de novo* strains, and the result with the highest similarity was chosen.

### RNA-seq quantification

RNA-seq reads were aligned to the assemblies using STAR (v2.7.10).[78] Reads from each strain were aligned twice, once using the strain's assembly and annotation as reference, and again using GRCm39 assembly and annotation as a reference. featureCounts (v2.0.1)[58] was then used to obtain gene counts matrices. DEseq2 (v1.34.0)[95] was used to detect differentially expressed genes with cutoff values of padj <0.05 and log2FoldChange ±1.

*RNA-seq read alignment using strain assembly*

```
STAR --runThreadN 16 --genomeDir '{strain}' --readFilesCommand zcat --readFilesIn SRR1030209_1.fastq.gz,
SRR1030209_2.fastq.gz,SRR1030210_2.fastq.gz --outFileNamePrefix '{strain}'_'{condition}'_1 --limitBAMsortRAM
38728510340 --outSAMunmapped Within --outSAMtype BAM SortedByCoordinate --outBAMcompression -1
```

*RNA-seq read alignment using GRCm39 assembly*

```
STAR --runThreadN 16 --genomeDir GRCm39_index --readFilesCommand zcat --readFilesIn '{strain}'_SRR1030201_1.
fastq.gz,'{strain}'_SRR1030202_1.fastq.gz  '{strain}'SRR1030201_2.fastq.gz,'{strain}'SRR1030202_2.fastq.gz
```

```
--outFileNamePrefix '{strain}'_'{condition}'_using_GRCm39 --limitBAMsortRAM 38728510340 --outSAMunmapped Within
--outSAMtype BAM SortedByCoordinate --outBAMcompression -1 --outTmpDir '{strain}'_'{condition}'_GRCm39
```

### Featurecounts for strains

```
featureCounts  '{strain}'_'{condition}'Aligned.sortedByCoord.out.bam  -a  '{strain}'_v3.5_1_19_X.gff3  -o
'{strain}'_'{condition}'_Counts -T 8 -t CDS -p -F GFF3 -g ID -O''
```

### Featurecounts for GRCm39

```
featureCounts  '{strain}'_'{condition}'_using_GRCm39Aligned.sortedByCoord.out.bam  -a  Mus_musculus.GRCm39.
104_1_19_X.gff3 -o '{strain}'_'{condition}'_using_GRCm39_Counts -T 8 -t CDS -p -F GFF3 -g ID -O
```

### Variable number tandem repeats (VNTR) in coding regions

The raw data and software source code are available on GitHub https://github.com/Oujiang-Laboratory-Bioinformatics/VNTR_discovery_by_Yolo.

In short, annotated CDS from GRCm39 reference and *de novo* assembly of 17 mouse strains are extracted with Gffread v0.12.7.[88] For the CDS shorter than 10Kb, equal length of ambiguous sequence (N) has been added on both ends. For the CDS longer than 10Kb, the sequences are split into parts, and the last split sequence is also completed with ambiguous sequences (N) if it is between 0.1 and 10kbp. Orthologues between GRCm39 and the strain genomes are processed into 750 x 750 pixel dot plot figures (830 x 830 pixel dot plot including the axis), with yass v1.16,[96] with parameters -M 3 -C 5,-4,-3,-4 -G −16,-4 -E 10 -X 30 -r 2 -d 1 -s 70. These dot plot figures are annotated by LabelImg v1.81 [https://github.com/HumanSignal/labelImg], using a rectangular box to annotate, and the output mode is in yolo format. In the process of labeling, the following criteria are uniformly adopted to determine if the signal blocks need to be annotated.

(1) Signal block with more than 5 repeats (show as parallel to the diagonal line on one side) with a clear outline should be annotated, for example (c.) (d.) (e.) below;
(2) The areas composed of multiple stable parallel lines with obvious gaps on the same image are regarded as different signal blocks, for example (f.) below;
(3) Block composed of ultra-short lines and points, with a not particularly clear outline should not be annotated.

After manual annotation, 262 samples are selected as the training and validation sets. About 80% of the total annotated figures were selected as the training set, and others as the validation set, while manual inspection ensured the same distribution of different signal blocks' patterns.

Model training is performed by yolov10 [https://github.com/THU-MIG/yolov10], with the command line 'yolo detect train data = mydata.yaml model = yolov10s.yaml'. The training process run for a total of 355 epochs, while the early stopping was implemented with a patience of 50 epochs. The performance of the model on the validation set is shown below.

| precision | recall | mAP50 | mAP50-95 |
|-----------|--------|-------|----------|
| 93.7% | 83.3% | 95.1% | 81.3% |

Pattern detection, using the trained model, is accomplished with the command line 'yolo detect predict model = runs/detect/train/weights/best.pt source=<file_folder> save_txt save_conf conf = 0.25 iou = 0.45'. The results are ranked by confidence. For all gene pairs with >0.80 confidence, the length change in tandem repeats is judged by the ratio of signal height and width (r). All records with r < 0.9 or r > 1.1 are further manually checked in IGV v2.18,[97] if raw PacBio reads aligned to GRCm39 have significant insertion/deletions. A summary of image segmentation can be found in Figure S9.

