## [Document S2. Transparent peer review records for Helmy et al. · Cell Genomics]

High quality mouse reference genomes reveal the structural complexity of the murine protein-coding landscape

Author list: Mohab Helmy, Jin U. Li, Xinyu F. Yan, Rachel K. Meade, Elizabeth Anderson, Patrick B. Chen, Anne M. Czechanski, Tomás Di Domenico, Jonathan Flint, Erik Garrison, Marco T.P. Gontijo, Andrea Guarracino, Leanne Haggerty, Edith Heard, Kerstin Howe, Narendra Meena, Fergal J. Martin, Eric A. Miska, Isabell Rall, Navin B. Ramakrishna, Alexandra Sapetschnig, Swati Sinha, Diandian Sun, Francesca F. Tricomi, Runjia Qu, Jonathan M. D. Wood, Tianzhen Wu, Dian J. Zhou, Laura Reinholdt, David J. Adams, Clare M. Smith, Jingtao Lilue, Thomas M. Keane

Summary

Initial submission: Received : May 29th 2025

Scientific editor: Judith Nicholson

First round of review: Number of reviewers: 2
Revision invited : June 24th 2025
Revision received : August 25th 2025

Second round of review: Number of reviewers: 1
Accepted : 28th October 2025

Data freely available: Yes

Code freely available: Yes

This transparent peer review record is not systematically proofread, type-set, or edited. Special characters, formatting, and equations may fail to render properly. Standard procedural text within the editor's letters has been deleted for the sake of brevity, but all official correspondence specific to the manuscript has been preserved.

Referees' reports, first round of review

Reviewer 1:

I was very excited to get this paper and I enjoyed reading it. It reads well and represents an important contribution to the field. However, it does have important limitations.

Major

1) We are on the precipice of having much better assemblies produced using ONT reads and newer software. Thus, it seems these assemblies, which have N50s that are much shorter than full chromosomes, will have short shelf lives since they will soon be replaced with T2T assemblies. Related to this, the paper needs to clarify that the 17 assemblies being presented are definitely not T2T, giving N50 stats in the abstract would be one way to do that.

1) The title emphasizes pangenome but unless I've misunderstood, most of the results are obtained using conventional genomes (17 linear genomes). For example, in the section "rediscovery of non-reference genes" I see the sentence "We extracted sequences for 196 from public databases or publications and aligned them to both GRCm39 and our de novo genome assemblies" which seems to indicate that this section was not using pangenome. I also think the MHC section isn't using pangenome. Indeed, as I read over the results, I'm not clear on whether any of the insights stem from use of the pangenome. I realize that tools for using the pangenome are still being developed, but the title and first sentence of the abstract are about pangenome...a promise that this paper doesn't deliver on. This makes me feel that the title and abstract are overselling the role of pangenome in this paper and should be changed. (Is this the first mouse pangenome? I thought that some publications had at least presented a pangenome for two strains: B6 and D2, but I perhaps those results are still unpublished.)

3) There are some omissions. For example, there is no section about short tandem repeats (as distinct from VNTRs) and also no section specifically

discussing structural variants. There is also no mention of the sex chromosomes or mitochondria. It seems that something should be said about each of these topics.

Minor

1) Move the contig and scaffold N50 info to table 1 (from sup table 1). This is related to my point about the near future where T2T assemblies will be more widely available. Discuss N50 stats / contig length in the first paragraph of the results. How do they compare to GRCh38?

2) There are usually regions of residual heterozygosity in inbred genomes. Some may actually be representative regions that are misunderstood, thus producing heterozygous calls even in an inbred strain due to an error in the reference. This idea comes up in a few parts of the paper, but I wanted to know how many apparently heterozygous regions remained in these assemblies or how het calls were handled. If that information was given, I missed it and those sorts of stats should be given more prominence.

3) A little more information about the "AI-based tool for VNTR identification" should be in the main text, at least enough that a reader can understand what steps are taken to assess the accuracy of this method. This is especially important because the authors portray the results of this approach as 'surprising'.

4) For the tuberculosis resistance vignette, did the authors try to determine the evolutionary/phylogenetic source of the insertion? Was it lost in B6 and other lab strains?

Very minor

1) The supplemental references section should be labeled as such, at one point I mistakenly thought it contained references for the main paper

2) Authors please verify that ref 51 discusses ADHD, T1D, and schizophrenia. (my institution is preventing me from accessing this paper). The abstract for reference 51 is not clearly focused on those phenotypes; this could be an (inadvertent) error?

Reviewer 2:

This is a highly consequential paper using long read sequence techniques on 17

inbred mouse genomes to create a pangenome assembly. They have revealed new genes, gene clusters, VNTRs, and assembly errors that will be invaluable for the mouse research community and will help with rigor and reproducibility of mouse studies. I have no major concerns or comments critical to the publication. It fits well with the journal and the data and its publication are crucial for genomics research in mouse field. I recommend publication without revision.

Authors' response to the first round of review

Dear Editor,

We would like to thank the reviewers for their thoughtful comments on our manuscript. Below we provide a point by point response of how we have actioned each reviewer comment. All changes to the original manuscript are denoted by red text.

Yours sincerely,

Thomas Keane (tk2@ebi.ac.uk), EMBL European Bioinformatics Institute

Jingtao Lilue (jli@ojlab.ac.cn), Oujiang Laboratory

Clare Smith (clare.m.smith@duke.edu), Duke University

Reviewers' Comments:

Reviewer #1

I was very excited to get this paper and I enjoyed reading it. It reads well and represents an important contribution to the field. However, it does have important limitations.

Major

1) We are on the precipice of having much better assemblies produced using ONT reads and newer software. Thus, it seems these assemblies, which have N50s that are much shorter than full chromosomes, will have short shelf lives since they will soon be replaced with T2T assemblies. Related to this, the paper needs to clarify that the 17 assemblies being presented are definitely not T2T, giving N50 stats in the abstract would be one way to do that.

We have added the contig N50 range to the opening sentence of the abstract and removed the pangenome term (next reviewer point):

"We present the first collection of 17 high-quality long read inbred mouse strain genomes with complete annotation (contig N50s ranging 0.8-33.9 Mbp)."

1) The title emphasizes pangenome but unless I've misunderstood, most of the results are obtained using conventional genomes (17 linear genomes). For example, in the section "rediscovery of non-reference genes" I see the sentence "We extracted sequences for 196 from public databases or publications and aligned them to both GRCm39 and our de novo genome assemblies" which seems to indicate that this section was not using pangenome. I also think the MHC section isn't using pangenome. Indeed, as I read over the results, I'm not clear on whether any of the insights stem from use of the pangenome. I realize that tools for using the pangenome are still being developed, but the title and first sentence of the abstract are about pangenome...a promise that this paper doesn't deliver on. This makes me feel that the title and abstract are overselling the role of pangenome in this paper and should be changed. (Is this the first mouse pangenome? I thought that some publications had at least presented a pangenome for two strains: B6 and D2, but I perhaps those results are still unpublished.)

In this paper, we are using the broad definition of a species pangenome being a set of high quality reference genomes from a species that capture the genetic diversity of a species or clade. A narrower definition related to using a specific pangenome data structure or tools to store and represent the genetic diversity of a species (e.g. minigraph, vg, panaroo etc.). It

should be noted that minigraph is used as the basis for figure 1 and 5 in our paper. As a compromise, we have changed the paper title to: "High quality mouse reference genomes reveal the structural complexity of the murine protein-coding landscape".

3) *There are some omissions. For example, there is no section about short tandem repeats (as distinct from VNTRs) and also no section specifically discussing structural variants. There is also no mention of the sex chromosomes or mitochondria. It seems that something should be said about each of these topics.*

We specifically did not include structural variation in our analysis as this topic was comprehensively addressed recently by Ferrar *et al.* (2023) to resolve genome-wide SVs in 20 genetically distinct inbred mice with long-read sequencing. A recent preprint by Ren *et al.* (2025) has comprehensively cataloged tandem repeats in 40 inbred mouse strains using long read technology. We have referenced both studies in the introduction: "In parallel, comprehensive genomic variation catalogs (SNPs, indels, tandem repeats, and structural variants) from dozens of strains¹⁴⁻¹⁷ and *de novo* assemblies of 16 key mouse strains have highlighted the extent of "high diversity" loci in the mouse genome¹⁸, revealing novel genes absent in the BL6 reference genome and mitochondrial genome variation¹⁸".

We sequenced female mice, and therefore do not have assembled Y chromosomes for the strains. A structural variation comparison of Chromosome X was included in the Ferrar *et al.* (2023) and Ren *et al.* (2025) studies.

A comparative analysis of mouse mitochondrial genomes was carried out in our 2018 paper (Lilue *et al.* (2018) Nature Genetics). We have added a sentence to the paper and reference: "...revealing novel genes absent in the BL6 reference genome and mitochondrial genome variation¹⁸."

Minor

1) *Move the contig and scaffold N50 info to table 1 (from sup table 1). This is related to my point about the near future where T2T assemblies will be more widely available. Discuss N50 stats / contig length in the first paragraph of the results. How do they compare to GRCm39?*

The contig and scaffold N50s for the strains and GRCm39 have been added to Table 1 and we added the following sentence to the first paragraph of the results:

"The contig N50s are lower than GRCm39 (0.82-33.9 Mbp vs. 57.4 Mbp for GRCm39), however the scaffold N50s are comparable to GRCm39 for several of the genomes (11-116 Mbp vs. 100.9 Mbp for GRCm39)."

2) *There are usually regions of residual heterozygosity in inbred genomes. Some may actually be representative regions that are misunderstood, thus producing heterozygous calls even in an inbred strain due to an error in the reference. This idea comes up in a few parts of the paper, but I wanted to know how many apparently heterozygous regions remained in these assemblies or how het calls were handled. If that information was given, I missed it and those sorts of stats should be given more prominence.*

We agree with the reviewer that residual heterozygosity in the inbred strains is an interesting and unanswered question. In our previous work, we noticed that hSNPs are primarily caused by errors in genome assemblies. For example, in 2018 we showed that for C57BL/6J reads aligned onto GRCm38, all hSNP rich regions are related to assembly mistakes in gene families, like SP140 family and Zfn members.

Below, we show the hSNP counts for every 1Mbp of genome using different CAST/EiJ assembly versions. The 2018 short read de novo (top) shows large amounts of hSNPs throughout the genome; these new long read de novo reduced more than 90% of hSNP, but still show hSNP rich regions (middle), and the T2T CAST/EiJ assembly (Francis *et al.* 2024, bioRxiv) has very few hSNP regions (bottom). A manual check of the hSNPs in the T2T assembly are coming from either micro satellites repeats, or remaining assembly mistakes.

Our conclusion is that there may be a very small number of 'true' hSNP interspersed in the genome, and they are likely coming from random mutations in the population. We suggest that hSNP analysis should be performed on future T2T level genomes, not from this set of genomes.

3) *A little more information about the "AI-based tool for VNTR identification" should be in the main text, at least enough that a reader can understand what steps are taken to assess the accuracy of this method. This is especially important because the authors portray the results of this approach as 'surprising'.*

We have added more information to the main text about the accuracy assessment that we had already carried out (details were in the materials and methods). We have added the following sentences to the main text:

"In a training and validation set consisting of 262 images (80% training, 20% validation), precision and recall were found to be 93.7% and 83.3%."

4) *For the tuberculosis resistance vignette, did the authors try to determine the evolutionary/phylogenetic source of the insertion? Was it lost in B6 and other lab strains?*

We looked at the genotype of the large inversion (assuming you mean inversion in the comment), and it appears to be also found in 129S1/SvlmJ. We have modified the sentence in the results noting this:

"Across the CC/DO founders, a non-syntenic region approximately 0.4 Mbp in length was identified on Chr15, containing an inversion present in BL6 and 129S1/SvlmJ and not found in the remaining seven founder lines (Figure 5e)."

Very minor

1) *The supplemental references section should be labeled as such, at one point I mistakenly thought it contained references for the main paper*

We have changed the heading of the supplemental references.

2) *Authors please verify that ref 51 discusses ADHD, T1D, and schizophrenia. (my institution is preventing me from accessing this paper). The abstract for reference 51 is not clearly focused on those phenotypes; this could be an (inadvertent) error?*

Thanks for pointing this out. Indeed, the reference is incorrect - we have removed it.

Reviewer #2:

This is a highly consequential paper using long read sequence techniques on 17 inbred mouse genomes to create a pangenome assembly. They have revealed new genes, gene clusters, VNTRs, and assembly errors that will be invaluable for the mouse research community and will help with rigor and reproducibility of mouse studies. I have no major concerns or comments critical to the publication. It fits well with the journal and the data and its publication are crucial for genomics research in mouse field. I recommend publication without revision.

We thank the reviewer for their high level of interest and enthusiasm for the paper.

Referees' report, second round of review

Reviewer 1:

The revised version addressed all my concerns.

Authors' response to the second round of review